# Digitalization as an Enabler to SMEs Implementing Lean-Green? A Systematic Review through the Topic Modelling Approach

**Geandra Alves Queiroz** [1,*] **, Paulo Nocera Alves Junior** [2] **and Isotilia Costa Melo** [2]

1  Engineering Department, Production Engineering, Universidade do Estado de Minas Gerais, Passos 37900-106, Minas-Gerais, Brazil
2  Escuela de Ingeniería de Coquimbo (EIC), Universidad Católica del Norte (UCN), Coquimbo 1781421, Chile
*  Correspondence: geandraqueiroz@gmail.com

**Abstract:** Small- and medium-sized enterprises (SMEs) play a fundamental role in the global economy. However, SMEs usually have different characteristics from larger enterprises, e.g., essential resource restrictions, lower performance, and higher environmental impacts. This requires them to search for strategies to be more competitive and sustainable. A possible solution relies on introducing Lean-Green practices. Previous research indicated that digitalization could be an enabler of Lean. Lean can also help to achieve increased environmental performance using the Lean-Green approach. In this study, this important yet under-studied area is investigated as we consider digitalization as an enabler for implementing lean in SMEs, with a focus on Lean-Green practices. A systematic literature review is executed, following a new framework based on topic modelling for extracting the papers. The topic modelling is executed through latent dirichlet allocation (LDA) which is a machine learning technique. In methodological means, this paper represents an example of the frontier of digitalization for research activities. Regarding the investigated focus, the main findings revealed that digitalization is an enabler to Lean and to Lean-Green. As digitalization supports information sharing, it consequently fosters performance measurement systems, improvements, and value chain integration.

**Keywords:** lean; lean-green; systematic literature review; topic modelling; industry 4.0; small and medium enterprise (SME); machine learning; artificial intelligence; digitization; digitalization; digital transformation; latent dirichlet allocation (LDA)

## 1. Introduction

Small- and medium-sized enterprises (SMEs) play a significant role in the global economy [1]. They face stiff competition in the globalized markets. Furthermore, these enterprises frequently present a lack of essential resources for organizational and technological changes as striking features [2]. They also tend to be less productive and pollute more than larger companies [3].

Remarkably, the Lean manufacturing (LM) paradigm is widely adopted in organizational strategy and is spread among several industries and service companies to improve productivity, quality, and delivery according to customer requirements by waste reduction and the minimum usage of resources [4]. In this context, to become more competitive, SMEs have tried to adopt the LM paradigm [5] and digital tools to improve the performance of their operations [6]. Digital tools are also fundamental pillars of the process of digital transformation (DT) which involves the combination of technology, organization models, and business processes aiming to create additional value for stakeholders within the changing digital environment [7].

Against this background, SMEs have been pushed to adapt their processes due to increased competitiveness in the manufacturing industry; in turn, they should embrace digital technologies and LM practices [8]. LM has become the essential approach for SME

performance because of the competitiveness, high levels of quality, low cost, delivery time, and flexibility [9]. It is important that SMEs look for agile and lean methods to reinforce their processes [10] because they are in an increasingly competitive environment where large companies are undergoing the new industrial revolution [11]. This industrial revolution is sometimes also known as digitalization, digitization, digital transformation (DT), or industry 4.0 [12]. Information and communication technology (ICT) tools enable SMEs to integrate their business process, and from that, SMEs improve their operational efficiency and productivity [12]. Furthermore, ICT is considered a valuable tool that collaborates with SMEs to achieve a competitive advantage as it can facilitate management decisions, e.g., the use of social media for customer engagement improvement [13].

Implementing industry 4.0 principles as digital tools is critical for business performance [14]. However, it has been emphasized [14] that SMEs must face more barriers to using new technologies than larger companies [14]. Consequently, this requires some specific procedures for SMEs. In the same line of reasoning, the effectuation of LM in SMEs has been discussed as an imperative requirement and a challenge in the operations management literature [5,9]. Approximately 10% or less of SMEs can implement lean after one year [15]; however, SMEs cannot use many aspects of Lean due to their size restrictions, flat hierarchies, and weak strategy alignment [2]. SMEs also present limited financial resources, fear of adopting new technology, lack of top management commitment, and poor leadership. These are the most critical barriers to LM implementation in SMEs in India, and SME managers view adopting new technologies as an expense rather than a strategic investment [9,16].

Additionally, in the globalization era, SMEs must ensure at the same time sustainable profitability through cost savings while being environmentally conscious [16]. Despite these barriers, the possible solution for sustainable (operational and environmental) improvements in SMEs could be introducing new technologies and approaches such as Lean-Green [17]. At one time, not all Lean practices were in line with Green strategies; Lean-Green represents the evolution of Lean towards sustainability. Alongside being more aligned with the sustainable goals of the current world, it was demonstrated that the vital link between operational excellence and Lean-Green enables the construction of competitive advantage in a sustainability era [18].

In this sense, digitalization can be an enabler to LM and Lean-Green in SMEs. At the same time, LM can be a good foundation for DT in SMEs [19], as the proposal of LM using the Lean tool value stream map (VSM) to support industry 4.0 implementation by the understanding of changes in materials, equipment, processes, and information flows [20]. Nonetheless, SMEs lack resources, and finding the best solution to implement the LM is difficult.

However, these studies have not addressed the relationship between digitalization and Lean implementation or Lean-Green in SMEs. Thus, it is necessary to understand how DT can catalyze Lean and Lean-Green and SMEs' competitiveness. The literature on the combination of LM and DT is relatively novel. Considering this, this paper aims to answer the following questions:

RQ1. How can digitalization support Lean implementation in SMEs?

RQ2. How can digitalization be an enabler in implementing Lean-Green in SMEs?

The purpose of this paper is to identify by theoretical discussion how digitalization can support Lean implementation in SMEs and verify whether it can facilitate the integration of Lean and Green in SMEs. Alongside the thematic relevance, this paper also represents a methodological contribution. An explorative systematic literature review is executed through topic modelling based on latent dirichlet allocation (LDA). LDA is a machine learning technique. This approach can be a novelty for explorative literature reviews [21,22]. Among the advantages, it can be highlighted that the categories and mapping do not need to be known in advance, coding can be automated, and costs and time of pre-analysis, analysis, and post-analysis are low [22].

This article is organized as follows. In the next section, more details about the main concepts are presented. In Section 3, the research method is detailed. The Section 4 brings the results that are discussed in Section 5. Finally, Section 6 brings the conclusions and suggestions for further works.

## 2. Background

### 2.1. Digital Transformation (DT)

The terms "digital transformation," "digitalization," and "digitization" have no broadly accepted definition, but they may be synonymous in many cases [23–25]. According to Gong and Ribieri (2021) [26], "Digital transformation is a fundamental change process, enabled by the innovative use of digital technologies accompanied by the strategic leverage of key resources and capabilities, aiming to radically improve an entity and redefine its value proposition for its stakeholders." This definition emphasises the importance of the transformation reaching a strategic level based on the available resources and capabilities. Innovation is also directly associated with the idea of digital technologies. An SME is the entity of interest in the current context.

Another term usually associated with this context is "industry 4.0" [27]. Although sometimes mentioned as a synonym of industry 4.0, according to Culot et al. [27], the concept of DT differs from industry 4.0 because DT stresses the implications for strategy and business model innovation, underlines the emerging technologies on the business model, and in turn, highlights the rise of cross-industry ecosystems. On the contrary, the term "industry 4.0" is mainly restricted to the DT process in the manufacturing sector. Similarly, "digitization" is often limited to the service sector [27].

Given this reason, the terms "industry 4.0" and "digitization" were not used in this literature search because the focus was not restricted to the manufacturing and service sectors. For the literature search, it was assumed that DT and digitalization were synonyms, similar to what was assumed by a previous paper [28]. In this regard, although the four terms ("digital transformation," "digitalization," "digitization," or "industry 4.0") were not all used in the literature search, they are used in this paper as synonymous and their meaning is the process defined by Gong and Ribieri (2021) [26] as a DT [28].

### 2.2. Lean Manufacturing and Lean-Green

Lean Manufacturing (LM) has been considered a manufacturing management paradigm. It is also recognized as the philosophy to manage organizations, englobing manufacturing, and services [4]. Liker and Morgan (2006) [29] argue that LM goals are the pursuit of better quality, shorter lead times, lower cost, higher employee morale, and better security to apply to any activity and industry. Jasti and Kodali (2014) [30] show that results from previous empirical works highlight that LM enables inventory reduction, quality improvement, and value addition for the customers at the right time.

In summary, LM is taken as a set of principles and practices to eliminate all forms of waste within an organization [31]. Shah and Ward (2007, p. 791) [32] define LM as "an integrated socio-technical system whose main objective is to eliminate waste by concurrently reducing or minimizing supplier, customer, and internal variability." These wastes are categorized into defects, inventory, inappropriate processing, overproduction, waiting, handling, transportation, and talent [33].

A complete list of Lean tools does not exist. Still, the most used are 5S, kaizen, value stream map (VSM), just in time (JIT), single minute exchange of die (SMED), total productive maintenance (TPM), kanban, standardized work, visual management, 5 why's (root cause analysis), and A3 report [32]. This list includes the management routine, pull production, information boards, and the overall equipment effectiveness (OEE) indicator [34].

LM philosophy has evolved as a successful instrument because of its significant impact on waste reduction, efficient management of resources, and the firm's performance [35]. Nowadays, companies are integrating areas and tools in Lean. The recent trend includes

the Green tools in Lean and digital tools or 4.0 in Lean. Worldwide, many organizations from the most diverse industries in the economy are embracing LM practices [36].

The Lean-Green has been considered an approach that supports the search for sustainable development of a production system's economic, environmental, and social pillars [37,38]. The Lean-Green focuses on waste reduction and the efficient use of resources [39,40]. Implementing Lean practices can offer significant advantages and synergies with a firm's environmental performance [37,41,42]. The empirical results from the study of Iranmanesh (2019) [43] show that Lean culture, involving process and equipment, product design, supplier relationships, and customer relationships positively and significantly affect sustainable performance. In the case of SMEs, Caldera, Desha, and Dawes (2019) [44] consider that the Lean-Green approach is an essential practice to support SMEs to achieve sustainability [44].

Additionally, digital tools have been discussed in the Lean literature as a very important dilemma in business [45]. The real-time information provided by digitalization is found to be very valuable in preparing an accurate VSM, which is the first stage in Lean implementation [46]. LM also offers great potential to implement innovative automation technologies in manufacturing [47], and digital tools can support manufacturing organizations to overcome the barriers of Lean implementation [48].

The literature shows that Lean and DT in SMEs can increase their competitiveness, exemplified by some Lean and digital tools. The study by Kolla, Minuferkr, and Plapper (2019) [49] shows that the combination of SMED "kaizen" and "heijunka" can help in SME product customization. As shown in a case study [50], in a food factory with the implementation of SMED, it was possible to achieve a 70% improvement in the overall OEE indicator. Digital tools, e.g., simulation software and 3D printing, can prototype and avoid the risk of product failure. This study also argues that ICT can support the integration of SMEs with their suppliers and customers. Furthermore, it can help to improve these relationships [49]. Another case resulted in improvements in the quality control process and Lean principles using the digitalization tools [51].

With regards to Lean-Green, the literature argues that it is essential to integrate the process information through the value chain [52,53] because the value chain integration is essential to the life cycle assessment (LCA) which contemplates a product life cycle "from the cradle to the grave." The information from the whole life cycle and value chain must be included in the environmental value stream map (E-VSM), a very important tool to integrate Lean and Green [53,54]. The proposal of E-VSM is that sustainability indicators, such as raw material consumption, water consumption, energy consumption, waste generation, and emissions, should be integrated into the traditional VSM [54,55]. The key issue for the implementation of Lean-Green is the inclusion of sustainability indicators in Lean systems [56,57]. Furthermore, this tool seeks to provide a wealth of information that can facilitate communication and management through Lean and Green indicators [56,57]. From this, the literature [3,18,54,56,57] argues about the importance of including Green performance indicators since this is what will allow integration with Lean systems and monitoring of environmental gains.

*2.3. Small and Medium Enterprises (SMEs)*

There is no globally standardized definition of small and medium enterprises (SMEs). The most common classifications are based on a financial measure and/or the number of employees, for example, the definitions from the European Union [58] and from South American countries such as Brazil [59] and Chile [60]. Some definitions, such as the Japanese, may specify whether the employees are temporary or permanent [61]. Even the same country may have different definitions of SME, depending on the sector, such as the USA [62]. In this study, it was assumed that all papers that used the acronym "SME" with the explanation "small and medium enterprises" and "small- and medium-sized enterprises" were referring to a comparable term. It is also considered that the term "SME"

could encompass micro (usually defined as enterprises with less than 10 employees) and self-employed enterprises.

SMEs are often associated with lower productivity. This is attributed to their inability to achieve economies of scale, difficulties they face in accessing credit or investment, lack of appropriate skills, and their informality [63]. There are also some characteristics that may affect the performance of SMEs, such as the industrial sector, management, technology, technical competence in marketing, and innovation [64]. One study demonstrates that innovation driven by digitalization plays an important role in mediating the relationships between resources for business model experimentation as well as business model strategy implementation and the overall SME performance [28]. Another study [8] suggests that SME owners and managers should allocate more importance to innovation and DT for achieving higher levels of organizational performance. Lastly, another study demonstrated that the availability of digital technologies is not sufficient for improving SME performance. It is also necessary to develop digital technology and relational and innovative capabilities [65].

Regarding the Lean practices in SMEs passing through DT, few papers have been written and the body of knowledge is not yet established. The main pillars of Lean practices originate from total quality management, just-in-time management, and time-based competition, as well as supply chain and activity-based management [66]. In this regard, DT is expected to foster Lean and consequently support sustainability [67]. One paper proposed a Lean-thinking framework for supporting the open data-based DT of SMEs in the forest service in Finland [66]. The authors concluded that Lean thinking was enabled through a digital platform where the needs and wishes of different stakeholders of the ecosystem were communicated [66]. In this regard, the current paper expects to contribute by identifying how digitalization can support Lean implementation in SMEs and verifying whether it can help/support the integration of Lean and Green in SMEs.

*2.4. Latent Dirichlet Allocation (LDA) Applied to Topic Modelling in SLRs*

There are many methods for conducting a systematic literature review (SLR). A previous taxonomy [68] compared five methods, i.e., reading, human coding, dictionaries, supervised learning, and topic model. Regarding the assumptions, reading and topic modelling do not require the categories, category nesting (if any), and mapping to be known. Different from reading, topic modelling can automate coding, and the relevant text features are known, i.e., reducing subjectivity. Literature review costs can be divided into pre-analysis, analysis, and post-analysis. According to the mentioned taxonomy, all methods require high post-analysis costs [68]. Comparing human (reading and human coding) to more automated literature review methods (dictionaries, supervised learning, and topic model), the analysis costs of the automated methods are lower, specifically the costs of person-hours spent per text and the costs of the level of substantive knowledge. A more specific advantage of topic modelling through LDA is the lower pre-analysis cost, specifically the cost of person-hours spent conceptualizing and the cost of the level of substantive knowledge, even when compared to other automated methods (dictionaries and supervised learning).

However, topic modelling is rarely applied in an exploratory literature review [21,22]. Among topic modelling methods, the machine learning technique of latent dirichlet allocation (LDA) is the most used and is considered state-of-the-art [69]. Asmussen and Møller (2019) [22] proposed a framework for researchers from any field to apply LDA for topic modelling in an explorative literature review. This framework was followed and applied here. The LDA is a probabilistic method that extracts topics from a collection of papers. A topic is a distribution of words over a fixed vocabulary. The semantics and meaning of the sentences are not evaluated. However, LDA analyzes the words in each paper and calculates the joint probability distribution between the observed (words in the paper) and the unobserved (the hidden structure of topics). According to an example provided by the framework's creators [22], whether one of the topics in a paper is Lean, it can be assumed

that the words Lean, just-in-time, and Kanban are more frequent compared to other papers that do not deal with Lean.

The LDA result is several topics, with the most prevalent topics grouped jointly. A probability for each paper is calculated for each topic, creating a matrix with the size of the number of topics multiplied by the number of papers [70]. While executing LDA, determining the number of topics (*k*) is a key parameter. As the framework proposed by Asmussen and Møller (2019) [22] is an unsupervised LDA, the relationship between the papers is not known before the model runs. Calculating the perplexity is normally used as cross-validation to estimate an adequate number of topics. Perplexity is a metric used to evaluate language models, where a low score indicates a better generalization. Lowering the perplexity is equivalent to maximizing the overall probability of papers being on a topic. Choosing the right number of topics is the art of balancing the right number of topics while keeping the perplexity at the lowest possible level.

## 3. Materials and Methods

This study adopted the combination of the SLR [71,72] and topic modelling through the LDA [22]. The primary phases of the SLR supported finding the studies that contained the keywords related to the generated topics. Alongside being beneficial to provide insights between the keywords and the topics, the topic modelling technique supported the paper's selection phase.

The SLR has been considered a replicable, scientific, and transparent literature review approach that minimizes bias. It is an iterative process used for identifying the extant literature about a research topic based on 5 steps: (i) RQs formulation; (ii) search strategy; (iii) selection and evaluation of relevant studies; (iv) analysis and synthesis of results; and (v) report the view [71]. The process framework used here is in Figure 1.

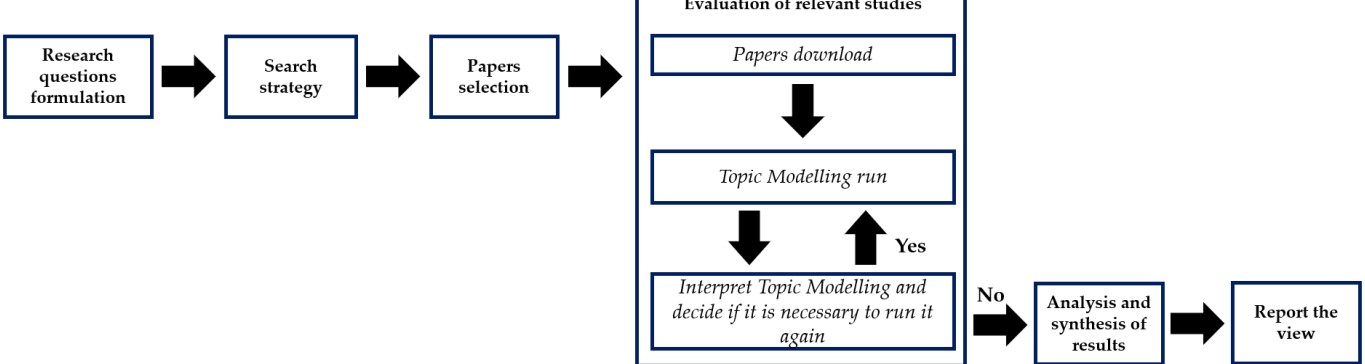

**Figure 1.** Schematic representation of the adopted framework, integrating topic modelling, based on latent dirichlet allocation (LDA), to the systematic literature review (SLR).

The first step defined the RQs, as presented in the introduction. Subsequently, the search strategy was designed and executed. Scopus and Web of Science were the databases chosen to find papers across reputable journals and conferences. These databases are regularly updated and have comprehensive international coverage [73]. It is important to emphasize that conference papers were selected because this research theme is new, so it is essential to capture the initial discussions held at scientific conferences [74]. The keywords were also identified among the 3 topics from a preliminary review on SMEs, Lean, and DT. The development of the search strings was done from the combination of the terms: small and medium enterprises (SMEs), digital transformation (DT), and Lean. The final search strings (available in Appendix A—the SLR Protocol) were defined after running tests to ensure reliable searches. After the search strings were established, the searches were undertaken in the databases. These searches resulted in 220 papers, but 47 of them were duplicated. They were excluded, resulting in 173 papers.

The step of paper evaluation (Figure 2) consisted of 3 phases for applying the topic modelling. The first phase consisted of the attempt to download the 173 resulting papers. Only 104 were available for download. The second phase consisted of the application of the topic modelling through LDA to perform the selection of the papers. One of the observed limitations of following the algorithm of Asmussen and Møller (2019) [22] is that the available algorithm is not able to read all papers. The reason why this happens is not clear. It is possibly due to some symbols and words non-identifiable by the algorithm. Topic modelling was applied to 92 papers, which the algorithm was able to read.

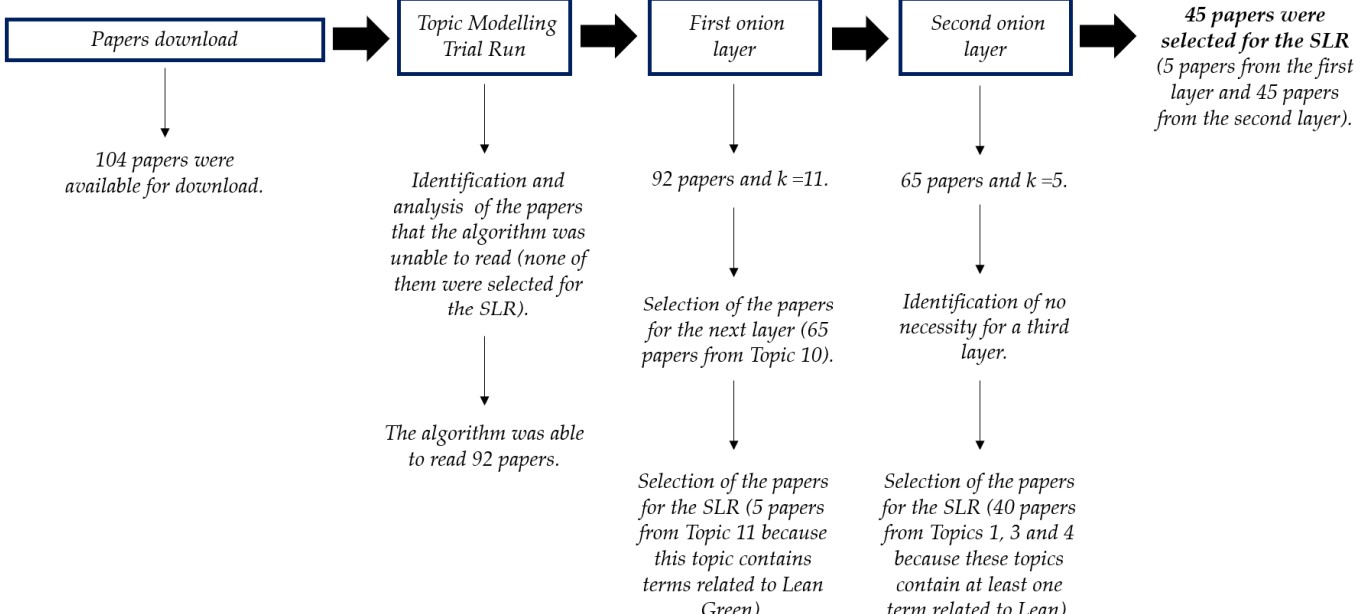

**Figure 2.** Flowchart of how topic modelling was applied for the selection of papers in the SLR and a detailed explanation of the number of selected papers per step.

The third phase of this step consisted of understanding the topic modelling results and judging whether it was necessary to run it again or not. It is expected that the larger the sample of paper, the more robust the results will be because of the lower perplexity levels. In general terms, 92 papers may be seen as a relatively small sample. Given this context, it was observed that above a certain threshold of topics (*k*), the LDA started to generate topics without any paper associated with it. Thus, it was decided to use the maximum number of topics (*k*) that still resulted in at least 1 paper associated with each topic. By observing the graphic of the number of topics (*k*) versus perplexity it was clear that, in the present case, the larger the *k*, the lower the perplexity. In other words, the model results had a better generalization.

As the objective is to use LDA for selecting papers, it is understood that the topics with fewer papers present a more discriminant list of words, which enables the analyst (a human being) to properly judge if these papers are pertinent to the RQ (or not). However, it was observed that in some cases, one topic has much more papers than the others. On the other hand, a topic with much more papers than the other will present a list of words that encompass many different themes, an obstacle to the adequate segregation of the papers. In such cases in this paper, the proposed solution is to run the LDA again, only with the papers on this topic, and to repeat this loop until the result does not present a concentration of papers on any topic. This is named the onion approach, and each loop is considered an onion layer. In the current case, it was necessary to perform 2 onion layers. Figure 2 details how this technique was used to filter out unwanted papers during the search for relevant literature. Further explanations are provided in the next section.

Finally, as is the case for any SLR, the final steps consist of analyzing and synthesizing the results (presented in Sections 4 and 5) and reporting the view on the theme (shown in the Discussion and summarised in the Conclusions).

## 4. Results

As presented in the previous section, the first onion layer had 92 papers. The maximum number of topics ($k$) that still resulted in at least one paper associated with each topic was $k$ = 11. The topics are presented in Table 1. Topic 10 has 65 papers, and the other 27 papers were distributed among 10 topics. Hence, the papers from Topic 10 were used for the second onion layer (Table 2). In the first onion layer, it was decided to select the papers associated with Topic 11 (five papers) for careful reading, interpretation, and discussion in the SLR. Topic 11 was selected because it presents the words "lean", "sme", "green", and "environment". In this way, it is the only topic that has words directly related to the three keywords. It is also the only topic with two words related to "Green" (Lean-Green) and is one of the focuses of the RQs.

**Table 1.** Topics generated in the first onion layer ($k$ = 11).

| T1 | T2 | T3 | T4 | T5 | T6 | T7 | T8 | T9 | T10 | T11 |
|----|----|----|----|----|----|----|----|----|-----|-----|
| lean | industri | servic | firm | data | var | vsm | suppli | learn | product | lean |
| implement | manufactur | construct | jit | erp | mainten | board | chain | construct | process | manufactur |
| startup | digit | agil | carbon | inform | cell | simul | manag | project | manag | smes |
| principl | factori | forest | adopt | bim | manufactur | director | strategi | train | system | manag |
| success | product | data | technolog | document | machin | audit | organis | plan | compani | implement |
| agre | procedia | softwar | malaysia | system | implement | govern | firm | cours | develop | perform |
| custom | autom | iot | cost | project | layout | twin | jit | technic | lean | factor |
| statement | smes | cloud | suppli | supplier | reduc | agent | technolog | elearn | research | green |
| disagre | logist | asset | relat | user | part | firm | supplier | educ | model | environment |
| compani | smart | transform | energi | order | engin | high | signific | engin | technolog | enabl |

**Table 2.** Topics generated in the second onion layer (i.e., $k$ = 5, considering the papers from the T10 from the previous layer).

| T1 | T2 | T3 | T4 | T5 |
|----|----|----|----|----|
| industri | design | lean | product | compani |
| product | system | manag | process | manag |
| manufactur | learn | implement | manufactur | technolog |
| digit | develop | smes | data | develop |
| smes | cost | product | time | inform |
| technolog | process | suppli | system | research |
| system | technic | manufactur | model | process |
| process | cell | chain | improv | model |
| lean | time | factor | vsm | project |
| custom | team | practic | sustain | andn |

In the second onion layer, the maximum number of topics ($k$) that still resulted in at least one paper associated with each topic was $k$ = 5. The topics are presented in Table 2. In this case, it was not observed the concentration of papers on one topic. It was therefore not necessary to perform a third onion layer. The papers from all topics that contained at least one word related to lean were selected for the SLR. The selected topics were 1, 3, and 4, and jointly they had 40 papers.

Given the discussed context, topic modelling through LDA supported the selection of 45 among 92 papers in an SLR (an extraction of around 49%), as shown in Figure 2. For achieving these results, there is no doubt that the time and cost invested were much smaller than those required when human beings execute the paper selection in an SLR. To guarantee that these faster results are reliable, measuring the perplexity is essential. However, it is also worth noting the empirical evidence of the current application. After

reading the 45 papers, only five of them were discharged from the SLR (11%) and they were all from the second onion layer.

Among the discharged papers, one had the acronym "SME" for structural modelling equation, and another had the same acronym for the Society of Manufacturing Engineers. The third paper was an editorial commenting on another paper. The fourth one did not consider DT, and the fifth did not consider Lean. Given what was exposed above, in this paper it is argued the level of error for the application of this framework is tolerable for an SLR, given the practicality, time, and cost savings. However, as with any methodological approach, it can be continuously improved. For example, the current machine learning technique (LDA) is unsupervised. Further research is recommended to investigate the possibilities of a supervised LDA as well as other machine learning techniques.

Finally, Tables 3–6 summarise the 40 papers selected and analyzed to answer the research questions of this study. These papers are manually separated into four groups according to the focus of the current study (Figure 3). Further developments of the algorithm are recommended to consider grouping, ranking, and cycling selected papers automatically.

**Table 3.** Summary of the papers that discuss Lean implementation in SMEs.

| Paper | Focus | Main Results |
|-------|-------|-------------|
| [15] | To provide and verify a step-by-step strategy from the low-to-high level of Lean application with appropriate tools and techniques to achieve the goals in each specific phase. | A roadmap for SMEs to successfully apply Lean in the context of limited resources. |
| [75] | This study identifies and prioritises critical success factors (CSF) based on implementing Lean Manufacturing for Thai SMEs. | "Technology resource" is the most important to enhance the implementation of Lean in SME organizations. |
| [76] | To identify the CSFs in managing maintenance (MM) activity in SMEs. | By identifying the CSF constructs, SMEs can utilize the information to improve their approach to a systematic MM program. |
| [77] | To provide an in-depth analysis of the implementation of Lean manufacturing in SMEs worldwide, to identify and present the critical difficulties that impact the undertaking, and to highlight the success factors of manufacturing firms. | The main barriers to Lean implementation are lack of leadership, commitment of top management, financial resources, resistance to change, training and knowledge about Lean tools, and know-how, skills, and expertise. |
| [1] | To provide a survey-based evaluation of applicability, benefits, and critical factors of Lean in SME product development and to identify how the Lean start-up approach operates in product development. | Lean methods can be introduced very quickly, promising high potential for improvements, and Lean methods combined with industry 4.0 technologies act as a booster for efficiency optimization in product development. |
| [78] | Based on the literature review, select the key success factors for Lean implementation in SMEs. These factors are then put in the context of industry 5.0 to explore the possibilities of Lean management as an enabler of industry 5.0. | Most of the success factors and tools are people-oriented, thus giving the human-centric approach to organizational and process improvement, as required by industry 5.0. |
| [79] | To identify key Lean practices for Indian automotive SMEs to reap the maximum benefits. | The findings show that customer involvement is the most important Lean practice, followed by problem identification and prevention, total productive maintenance, and others. |
| [80] | To identify and suggest improvements in two areas crucial for implementing Lean principles in an aluminium semi-permanent casting small to medium enterprise (SME). | This paper builds an analytical model to measure production performance and presents a case study of an aluminium foundry. The case study reveals that using two Lean principles of bottleneck identification and Lean buffering could allow for a timely response to customer requirements. |
| [81] | To explore how Lean principles can improve the entrepreneurial ecosystem in Malaysian SMEs. | Using Lean principles could be more cost-effective because they avoid lengthy implementations of unproven strategies and investments in the entrepreneurial ecosystem. |
| [82] | To identify the steps SMEs can follow in searching for a plant distribution model and applying layout improvement to increase productivity. | Implementing the tools for plant distribution and 5S was proposed to reduce waste in the production process since this generated a high total cycle time. |

**Table 3.** *Cont.*

| Paper | Focus | Main Results |
|-------|-------|--------------|
| [83] | To analyse which aspects of the JIT philosophy also apply to small- and medium-sized enterprises (SMEs). | Barriers include a lack of supplier cooperation and partnerships, an inability to develop the necessary technologies and methodologies to reduce or eliminate waste, difficulties in managing demand fluctuations, a lack of capital to acquire advanced technologies, quality control problems, and inadequate employee training and development. Enablers include the ability to empower employees, reduce JIT implementation time, overcome employee resistance to change, and receive various forms of governmental support. |
| [84] | To investigate the opposing factors in adopting Lean manufacturing in Indian SMEs and systematically evaluate causal/effect barriers by the Grey-DEMATEL technique. | Nine out of fourteen barriers belong to the casual group, and five barriers belong to the effect group. This study revealed that fear of adopting new technology is a high influencing barrier among all barriers. |
| [9] | To investigate the interrelationships among LM adoption barriers in Indian SMEs. | The findings show that limited financial resources, fear of adopting new technology, lack of top management commitment, and poor leadership quality are the most critical barriers to LM diffusion in Indian SMEs. |
| [85] | To analyse the contextual relationship and dependency amongst enablers for LM implementation in Bulgarian SMEs. | The findings demonstrated that "leadership and commitment by management," "human resource management," "customer relation management," "supplier relation management," and "information technology system" are the most significant enablers for Lean implementation in Bulgarian SMEs. |

**Table 4.** Summary of the papers that discuss the integration of DT and Lean.

| Paper | Focus | Main Results |
|-------|-------|--------------|
| [20] | To explore the integration of the VSM with simulation to help industry 4.0 initiatives. | VSM combined with simulation can help enterprises understand changes in materials, equipment, processes, and information flows associated with industry 4.0 application scenarios. |
| [86] | To understand how effective Turkish SMEs use enterprise resource planning (ERP) systems to assess their adherence to Lean. | The effective usage of specific ERP modules can contribute toward applying Lean principles and vice versa. |
| [11] | To propose a digital twin-enabled VSM approach for SMEs. | A theoretical framework which combines IoT data-driven production process planning and simulation methods. |
| [87] | To propose that entrepreneurial orientation (EO), participative management style, supplier relations, resource management, JIT, and technology utilization are several drivers of an effective management decision-making approach. | EO, supplier relations, resource management, JIT methodology, and TU positively impact agile supply chain management. |
| [45] | To understand how Lean and industry 4.0 integration can contribute to increased flexibility and productivity. | Implementation of LP and industry 4.0 concepts in Angola are not being uniformed in terms of company size and geography. Five S is the leanest tool implemented. |
| [88] | To investigate business intelligence (BI) tools that help an SME to improve its supplier order fulfilment management. | SMEs can benefit from BI tools at an affordable cost, joining the emergent trend of deploying business analytics to make systematic innovations and collaborating with supplier relationships to implement a just-in-time strategy. |
| [7] | To discuss the problem of digital transformation of supply chain management (SCM) due to the changing business environment and the desire to meet customer demand in Thailand. | SCM processes are altered, such as reduction in product design and a manufacturing period, faster delivery of products to customers, easily meeting of the preferences and demands of customers, and faster and effective decision-making supported by big data and analytical decision techniques. |
| [49] | To provide state-of-the-art literature on existing assessment models and consequently map Lean and industry 4.0 components to the specific characteristics of manufacturing SMEs. | Development of a hybrid model including Lean and industry 4.0 features suitable for manufacturing SMEs. |

**Table 4.** *Cont.*

| Paper | Focus | Main Results |
|---|---|---|
| [89] | To document the positive experience of one enterprise in jointly auditing and improving data quality and IT infrastructure, which better aligned its XPS with sustainability objectives. | The paper provides a management supporting tool to promote change in the natural evolution of businesses, for example, the BI panels. Regarding the link between sustainability and business objectives via big data and BI tools, the company detected one weakness concerning data analysis. |
| [19] | To demonstrate the state-of-art regarding Lean production and digitization and to present an approach based on the consistent opinion of the reviewed literature, which formulates digitization as the next step of Lean management in production systems. | This study reveals the demand for a methodological approach in an SME environment that quantifies the profitability of implementing digital technologies in Lean. It formulates digitization as the next step of Lean in production systems. |
| [90] | This paper aims to understand the impact of digitalization on international, Lean, and global start-up speeds (LGS). | Digitalization allows LGS to increase decision-making efficiency and optimize international market evaluation strategies and processes. |
| [14] | This paper presents the value stream mapping analysis results supported by the overall equipment effectiveness (OEE) coefficient analysis. | It was found that the partial digitization of the one production operations in SMEs has a positive effect on the course of the process. Creating a running chart for the variability of Total Productive Maintenance (TPM) indicators helps to improve the organization and efficiency of work. |
| [91] | To present the outcomes of expert workshops to identify requirements of SMEs in the field of smart logistics management. | SMEs will only benefit from industry 4.0 by following customized implementation strategies, approaches, concepts, and technological solutions. |
| [92] | To evaluate how Lean can support industry 4.0 in pursuit of greater customer value and manufacturing excellence. | This paper finds that the pursuit of SMEs toward process efficiency and waste reduction can be best achieved through a focus on foundational digitalization and data management, then taking a stepwise approach towards the cyber-physical systems of industry 4.0. |
| [93] | To investigate Lean manufacturing and healthcare logistics 4.0 concepts, methods, and tools. | The framework elaborated for implementing industry 4.0 and logistics 4.0 concepts in SMEs, in opposition to many architectures of the literature, considers sustainability as the kernel of industry 4.0 concept implementation in SMEs. |
| [94] | To examine driving factors of Lean supply chain management and major supply chain and information technology solutions applied in SMEs. | There is a positive relationship between the successfulness of Lean supply chain practices and IT solutions adopted, and the Lean supply chain would increase. It is also shown that these supply chain and information technologies play crucial roles in helping small companies transform into leaner organizations. |
| [95] | To present a case study about a Chinese company that constructed an intelligent Lean system. | China's small- and medium-sized manufacturing enterprises can promote intelligent Lean through the path of "standardization, lean, digital, intelligent" concepts. |
| [96] | To investigate the relationship between Logistics 4.0 concepts and technologies and logistics performance indicators in manufacturing enterprises. | The implementation of smart and Lean concepts has a major impact on logistics performance, whereas information and communication technologies, as well as autonomous logistics systems and vehicles, are not completely implemented and exploited yet. |
| [12] | To analyze the relationship between information and communication technologies (ICT), industry 4.0, and agile manufacturing. | ICT plays a key role, but it is not a goal itself. They are a prerequisite for the implementation of industry 4.0. Still, they serve to achieve agility in the manufacturing system and, as a result, achieve a competitive advantage for enterprises operating in turbulent and unpredictable environments. |

**Table 5.** Critical success factors (CSFs) to DT adoption in Lean SMEs.

| Paper | Focus | Main Results |
|---|---|---|
| [97] | A theoretical association of advances in manufacturing technology and tools. | Framework with the technologies and process improvement methodologies to point out the difficulties of SMEs to obtain technological advancements because of limited resources and lack of finances available to them. |
| [66] | To study the functional areas which can potentially leverage industry 4.0 technologies and help India's SMEs to adopt digital technologies for the identified functional areas. | Manufacturers want to change their design and manufacturing strategies based on performance metrics. Therefore, they need first to capture real-time machine data, analyze it, and then incorporate the resulting improvements in manufacturing and design decisions in that order. |
| [98] | In this paper, the analysis of the SMEs, and automation integrators of the project, identify a correlation between the challenges, age, and size of the SMEs. | It is presented that a strategic focus on production with "simple" smart technology concepts can enable SMEs to become more adaptable to the changing and dynamic environment. For example, collaborative robots and AGVs (automatic guided vehicles) in a reconfigurable environment can adapt to changing environments. |
| [99] | To show global challenges, such as digital transformation, are urging companies to become more dynamic and flexible. | The Lean start-up approach in large manufacturing companies examined the feasibility of accelerating product development while it makes innovation processes cheaper, more flexible, and more reliable. |

**Table 6.** DT and Lean-Green.

| Paper | Focus | Main Results |
|---|---|---|
| [100] | This paper proposes a method to drive process innovation toward the increase in efficiency of a production plant. | Correct data management permits to plan the best practices to improve processes and systems involved, in terms of environmental and economic impacts, meaning a process of sustainable innovation. |
| [17] | To analyse the Lean-Green performance of Indian manufacturing SMEs by investigating the influential relationships of various factors and the firm's set of Lean and Green practices. | Enterprises need to decrease their operational sizes to improve operational and environmental performance. The possible alternative and more practical strategy could be introducing new technology innovation and holistic adoption of manufacturing excellence initiatives, such as Lean-Green. |
| [16] | To identify the drivers for integrated Lean and Green manufacturing from the combined support of existing literature and expert opinions. | The results reveal that top management commitment, technology up-gradation, current legislation, green brand image, and future legislation are the five most important drivers for implementing integrated Lean-Green in Indian manufacturing SMEs. |

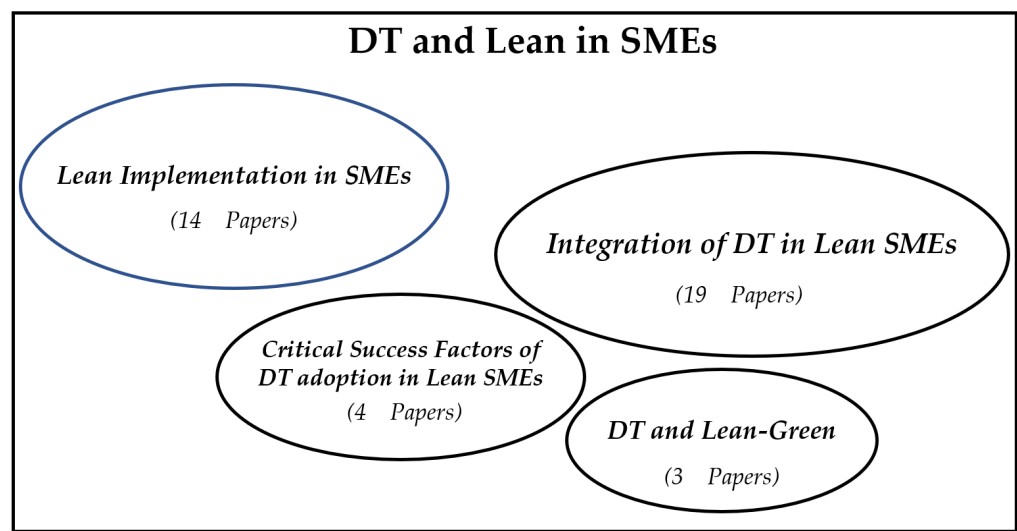

**Figure 3.** Conceptual representation of the results.

Tables 3–6 present the objective (focus) and the main results of each group of papers shown in Figure 3. Table 3 is the group of 14 papers that discuss Lean implementation in SMEs. Different from the papers in other groups, these papers bring DT as a remarkable secondary point but not as the focus of the investigation. Table 4 brings the 19 papers that focus on DT in Lean SMEs. Table 5 brings four papers that discuss the critical success factors (CSFs) of DT adoption in Lean SMEs. Finally, Table 6 brings three papers that discuss DT and Lean-Green in SMEs, which is also one of the RQs of the current SLR.

## 5. Discussion

### 5.1. How Can Digitalization Support Lean Implementation in SMEs?

From these 40 studies, it was possible to answer the RQs and find some propositions about Lean, SMEs, and Digitalization. The first point observed from the studies is that DT can collaborate on information sharing, which is essential for a successful Lean implementation. The lack of resources for advanced technologies can hinder Lean in SMEs. For example, the information technology system was highlighted as one of the most important facilitators for implementing Lean in Bulgarian SMEs [85]. The findings indicated that information sharing from digitalization could be the enabler, mainly for VSM, JIT, Lean Supply Chain, performance indicators (e.g., OEE), TPM program, and ERP systems.

The VSM tool needs information from the process, such as lead-time, efficiency cycle time, etc., to analyse and improve the process [101]. This is exemplified by a case [14] in which the partial digitalization of the operations of an SME helped to visualize the efficiency data measured by the OEE. These data helped in the construction and visualization of the current situation in the VSM. Thus, it was possible to trace a VSM of the future situation based on concrete data, which led to satisfactory results of efficiency increase in availability, performance, and quality. At the same time, the information flow achieved through VSM can assist the implementation of technologies from industry 4.0, entering a virtuous cycle where digitalization fosters Lean practices and vice versa [20].

Lu, Liu, and Min (2021) [11] detail the integration of DT in the VSM in production line optimization. This study shows that from digital information through a radio frequency identification (RFID) chip, a new VSM solves the problems where the standard VSM method is powerless. The RFID reader enabled managers to give directly quantized answers to the problems and to be notified about other possible circumstance changes and problems after each implemented optimization plan, e.g., the daily output capacity, average takt and reasonable work in progress (WIP) inventory, etc. Therefore, these data are a timestamp table which allows the managers to see every change in the process in a faster way and make more accurate calculations for the implementation of improvements, such as a Takt matrix, which can be used for modelling and making an overall process simulation of mixed series production.

The case of JIT paradigm requires suitable technologies that allow information sharing to adequately plan the flow of materials between the production chain. As exemplified in [87], the process information in SMEs informing JIT supplier partnerships aim towards order fulfilment cycle time reduction. Another example is shown through empirical results from the implementation of Lean integrated with concepts of industry 4.0 or digitalization. The integration results in a higher goods security level and a real-time data sharing policy [102]. Therefore, there is a positive correlation between the effectiveness of Lean supply chain practices, such as JIT, and the level of information technology solutions implemented. This is also essential to help SMEs implement Lean [93]. Information and communication technologies and Lean implementation have a great impact on logistics performance [95]. In agreement with these studies, in the study of Dowlatshahi and Taham (2009) [83], the hypothesis arises that advanced and relevant manufacturing technologies and methodologies are necessary for enhancing JIT implementation in SMEs.

The research of Gandhi, Thanki, and Thakkar (2019) [16] argues that SMEs present an environment with high uncertainty and low flexibility, so, from this aspect, the adoption of internet and communication technology can provide a competitive advantage. This is

because, as pointed out by Alshamaila et al. (2013) [103], if SMEs adopt cloud computing technology they can receive benefits in terms of cost-saving, access to a large pool of hardware resources, enhanced information sharing, improved supply chain collaboration, and a faster time to market. Another finding relates to the use of ERP. The research [74] pointed out the effective use of some ERP modules collaborated to implement Lean.

The use of a great amount of data in ERP systems can provide support to build collaborative supplier relationships, which is the foundation of JIT in SMEs [87]. However, SMEs are often unable to acquire the advanced technologies needed for the effective implementation of JIT due to limited financial resources [83], but it was found that BI tools can be used at affordable costs [87].

Alongside Lean supply chain practices, other Lean practices can be supported by DT tools in SMEs. Some studies show that DT tools in SMEs can have a positive impact on TPM, visual management, and performance indicators [14,76]. The data for the TPM indicators collaborate to visualize the variability and motivate the workers to improve the process efficiency. Therefore, they help to visualize the process information and are a good way to implement new management concepts in SMEs [14,76]. These cases [76] show that visualization in maintenance management collaborates to improve productivity, obtaining superior product quality, optimizing operation costs, improving the delivery of products to customers, and empowering employees. A survey of Indian SMEs indicated that manufacturers that aim to make some improvements in their operations must capture real-time machine data, analyze them, and incorporate the resulting improvements in manufacturing and design decisions in this order [66].

Another critical issue is that the DT allows Lean to optimize the decision-making process contributing to internationalization as the international managers work to integrate forces and mediate between local market realities and corporate goals [89]. Another consequence of increasing information sharing in SMEs through DT tool implementation is supply chain collaboration, because it is vital to place SMEs in the current context of market trends and the needs of their stakeholders [7]. In short, based on the results of this literature review, it is clear that information is the main way in which digital tools collaborate for Lean in SMEs.

Furthermore, it was found that the use of simulation tools supports the viability of Lean improvements in SMEs. It was demonstrated [82] that layout changing and 5S reduce waste in the production process. Simulation software to test and validate the improvements before the implementation is most effective in reducing the total cycle time. It can also reduce the risk and help to find better solutions. Related to the first RQ, it was also found that the Lean start-up approach can help SMEs to accelerate the innovation process and makes it cheaper, more flexible, and more reliable despite the smaller resources than for large companies [10,99].

In short, based on the results of this literature review, it is possible to understand that digital technologies can be an enabler of Lean implementation. Firstly, it is clear that information sharing from digitalization is the main way in which digital tools collaborate for Lean in SMEs. It was noted that for making decisions or implementing improvements, it is necessary to visualize the data. In this case, to Lean work properly, collaborative supplier relationships are essential. Technologies such as simulation software also allow the process to better understand and reduce project risks.

*5.2. How Can Digitalization Support Lean-Green Implementation in SMEs?*

This RQ is quite unexplored in the literature, but it was possible to obtain some findings/deductions. Similar to the first RQ, information sharing from digital tools is the central method in enabling the Lean-Green in SMEs. The study of Papetti et al. (2016) [99] argues that correct data management allows the best practice implementations to improve sustainable innovation of the operations, that is, in terms of environmental and economic impacts.

It is essential to highlight that process and supply chain information is more critical when environmental impacts must be considered [52,53,55]. This information allows the data to properly allocate consumption and, consequently, identifies the causes of a specific consumption trend (e.g., under/overproduction, increase in energy consumption, etc.) [99]. The study on Indian manufacturing SMEs found that technology up-gradation (utilization of energy resource efficiency advanced technology) is one of the five most important drivers behind Lean and Green manufacturing integration [16].

A case study driven by a building material supplier located in Hong Kong showed that data provided by BI tools helped the SME to detect weaknesses in data analysis and allocated sustainability-specific data to improve the efficiency of the entire production unit. The method used is the basis for a web application tool [87]. Another example of process data was demonstrated in a study that proposed a method to increase the efficiency of a production plant in terms of energy consumption [99]. These cases are in line with the Lean-Green integration proposals that highlight the importance of including sustainability indicators. In this instance, it is necessary to have access to process information. Another issue related to Lean-Green is the increase in their sustainable performance. It has been demonstrated that collaborations with big companies and investing in technology improved communication between employees and suppliers. It can also reduce process lead times and achieve better operational performance [17].

On these grounds, it is possible to infer that DT can collaborate with Lean-Green in a similar but more relevant way than Lean. This proposition is supported by the life cycle approach of environmental aspects. As understood by the Lean-Green literature which shows the direct effect that information sharing has on Lean decisions, Lean-Green needs even more integration of the data from the whole value chain. In this way, the environmental impact data from all value chains and during the product life cycle are the foundation for sustainable improvements. Thus, when digitalization enables the Lean-Green implementation in SMEs, these firms can obtain more advantages over their competitors and collaborate for sustainable global targets since they are more unproductive and more pollutant than larger companies.

## 6. Conclusions

This paper aimed to investigate the relationship between the DT in the Lean and Lean-Green in SMEs using a topic modelling systematic review approach. With this approach, it was possible to select 40 papers and find theoretical evidence fast, effectively, and at a low cost, proving that this approach is a viable option for the realization of SLRs. Thus, the method helped to find the state-of-art effectively and from a vast number of papers and to understand relationships among the studied constructs. Nowadays, it is possible to find many papers in databases, and the time for paper selection is usually the limiting factor for completing SLRs.

Based on the findings, it was possible to understand that the DT can enable Lean and Lean-Green, and that both can improve SME competitiveness. The main point highlighted in the studies was the information sharing obtained from the DT. The results showed that the integration and visualization of the process data collaborate with the performance measurement system, helping to make decisions. It was also demonstrated that the implementation of the tools JIT, ERP, VSM, TPM, and E-VSM could be better supported by the DT, which provides the process data. In the case of JIT and E-VSM, the DT provides the information from the value chain. Therefore, regardless of Lean-Green, obtaining more data from the value chain is emphasized as an essential requirement for sustainable improvement. Despite some relationships found in the literature, this area still requires more research, mainly regarding the DT and Lean-Green in SMEs. Thus, future studies are recommended to quantitatively investigate how much Lean-Green and DT implementation improve SME sustainability performance.

As with all research, the current study is not free from limitations. It is necessary to consider that some papers were unavailable for download. Thus, some contributions

from these unavailable works may have been missed. The adopted algorithm could not read all papers and it should be improved to enable the reading of all. Another limitation refers to the fact that only one filter could be replaced by the automatic solution. Further development of this framework could directly search the papers, check for duplicated ones, exclude them, and download the remaining papers. Alongside solving these issues, the algorithm should be able to read some two- and three-word expressions as one word for generating a topic. For example, the expression "supply chain" would be considered as two words "suppl" and "chain" in a topic by the current algorithm. It should be considered "supply chain" as one word in a topic. Further developments of the algorithm should also consider integrating visualization tools, such as a cloud of words, ranking, or cycling, enabling a faster interpretation of the results.

Finally, it is important to mention that the selection was based on an unsupervised LDA, which implies that the selected papers might not always be the same for all times the model is running. Therefore, the findings here represent a theoretical approach, and any generalization must be cautious. For this reason, other automatic techniques should be tested, and the differences between the results should be investigated.

**Author Contributions:** Conceptualization, G.A.Q., P.N.A.J. and I.C.M.; methodology, P.N.A.J. and I.C.M.; validation, G.A.Q. and I.C.M.; formal analysis, G.A.Q. and I.C.M.; investigation, G.A.Q. and I.C.M.; resources, G.A.Q. and I.C.M.; writing—original draft preparation, G.A.Q. and I.C.M.; writing—review and editing, G.A.Q., P.N.A.J. and I.C.M.; funding acquisition, P.N.A.J. All authors have read and agreed to the published version of the manuscript.

**Funding:** The Universidad Católica del Norte (UCN) partially supported the payment of the Article Processing Charge.

**Institutional Review Board Statement:** Not applicable.

**Informed Consent Statement:** Not applicable.

**Data Availability Statement:** Not applicable.

**Conflicts of Interest:** The authors declare no conflict of interest.

## Appendix A. SLR Protocol

| Literature Review Protocol | |
|---|---|
| Objective | To understand how digitalization enables Lean in SMEs. |
| Research Question | RQ1. How can digitalization support Lean implementation in SMEs? <br> RQ2. How can digitalization be an enabler in implementing Lean-Green in SMEs? |
| Keywords and Synonyms | digital transformation; digital transition; digital innovation; digitalization; industry 4.0; SME; small- and medium-sized enterprises; digital media; social media; social network; internet; technology; ICT; small and middle compan*; small and middle firm; lean; Toyota production system; just in time. |
| Source Selection Criteria Definition | Criteria: The sources should be available and globally recognized as high-quality sources. <br> Studies Language: English. <br> Source Search Methods: The sources should be available and globally recognized as high-quality sources. <br> Source List: Web of Sciences and Scopus. |
| Studies Type Definition | Research published in journals, books, and conferences. |
| Studies Initial Selection: | 3 August 2022. |
| Studies Quality Evaluation: | The quality is defined by the databases selected. |

| Literature Review Protocol | |
| --- | --- |
| Search string | Web of Science (38 papers returned): <br> TS = ((((((("digital transformation" or "digital transition" or "digitalization" or "industry 4.0" or "digital media" or "social media" or "social network" or "internet" or "digital" or "technology" or "ICT") and ("lean" OR "toyota production system" OR "Just in time ") and ("sme" or "small and medium firm" or "small and medium compan*" or "small and medium-sized enterprise")))))) <br> Scoupus (182 papers returned): <br> TITLE-ABS-KEY ((("digital transformation" OR "digital transition" OR "digitalization" OR "industry 4.0" OR "digital media" OR "social media" OR "social network" OR "internet" OR "digital" OR "technology" OR "ICT" ) AND ("lean" OR "toyota production system" OR "Just in time") AND ( "sme" OR "small and medium firm" OR "small and medium compan*" OR "small and medium-sized enterprise"))) |
| Data Extraction Form Fields | Filter 1—Exclusion of duplicate studies. <br> Filter 2—Download of papers. <br> Filter 3—Topic modelling application for papers extraction. |

| | Inclusion Criteria | Exclusion Criteria |
| --- | --- | --- |
| Study Selection Criteria (Filter 4), executed through human reading | • The paper addresses some aspects of digital transformation and Lean in SMEs. <br> • The paper addresses some aspects of digital transformation and Lean-Green in SMEs. | • The paper does not consider digital transformation. <br> • The paper does not consider SMEs. <br> • The paper does not consider any Lean practice. <br> • The paper is not in English (only the abstract is in English). |
| Studies analysis | Forty studies were analyzed, focusing on answering the RQs. | |

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
