# Peer review of "Digitalization as an Enabler to SMEs Implementing Lean-Green? A Systematic Review through the Topic Modelling Approach"

_sustainability, doi:10.3390/su142114089_

Round 1

Reviewer 1 Report

Dear Authors

Please see my comments that might help you to improve your manuscript

1) Line 37 - Costumer (spelling error) - Please go through whole manuscript to see if there are similar errors.

2) Line 57 - "As it is discussed in Section 2...". You cannot ask readers to read section 2 while in Section 1. 

3) Following articles might be relevant :  https://doi.org/10.3390/su132212341; https://doi.org/10.3390/su14148279

4) Section 2.1 would be more effective if you can explain terminologies digitization, digitalization and digital transformation.

5) Material and methods - You have used SLR, it would be beneficial if you use a flowchart to show how you filtered out unwanted literature during search for relevant papers.

6) Table 3 needs to be presented in a better manner as it is 12 page long.

7) Conclusion section needs to be tightened and reflect only on outcome and author views, drawbacks and future recommendation.

regards  

Reviewer 2 Report

The article is devoted to an actual topic. Careful production is becoming more and more dominant factor of sustainability every day.  The authors use proven research methods, they have obtained important results based on the comparison of definitions of DT. However, the authors claim that the results showed that the integration and visualization of process data interact with the performance measurement system, helping to make decisions. But unfortunately, the article does not present the results of visualization. In addition, the fable 3 Summary of the 40 papers selected and analysis for answering the Race in this SLR takes 11 pages, which is quite inconvenient to read. Just in this case, data visualization (ranking or cycle, etc.) would help the authors.
